# Vegetation Recovery Patterns in Burned Areas Assessed with Landsat 8 OLI Imagery and Environmental Biophysical Data

**Bruno M. Meneses** 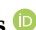

Centre for Geographical Studies and Associated Laboratory TERRA, Institute of Geography and Spatial Planning, Universidade de Lisboa, Edif. IGOT, Rua Branca Edmée Marques, 1600-276 Lisboa, Portugal; bmeneses@campus.ul.pt

**Abstract:** Vegetation recovery after the large wildfires that occurred in central Portugal in 2017 is assessed in the present study. These wildfires had catastrophic consequences, among which were human losses and a vast extent of forest devastation. Landsat 8 OLI images were used to obtain the land use and cover (LUC) classification and to determine the Normalized Burned Ratio index (NBR) for different times. NBR results were used to determine the difference between the NBR (dNBR) before the fire (pre-fire) and after the fire (post-fire), and the results obtained were cross-checked with the LUC. The dNBR results were cross-referenced with biophysical data to identify the characteristics of the most important burned areas in need of vegetative recovery. The results showed the spatial differentiation in vegetation recovery, highlighting different factors in this process, in particular the type of vegetation (the predominant species and bank of seeds available), the biophysical characteristics of burned areas (for example, the soil type in burned areas), the continentality gradient, and the climate conditions. The vegetation recovery was differentiated by time according to the species present in the burned areas pre-fire. In general, shrubland recovery was faster than that of tree species, and the recovery was more marked for species that were regenerated by the rhizomes after fire. The recovery process was also influenced by the season in the study area. It was more efficient in the spring and at the beginning of the summer, highlighting the importance of optimal conditions needed for vegetation regeneration, such as the temperature and precipitation (soil humidity and water availability for growing plants). The results of this research are important to forest planning: the definition of the strategies for the ecosystems' recovery, the adoption of preventive measures to avoid the occurrence of large wildfires, the modification of anthropogenic practices, etc.

**Keywords:** vegetation recovery; regeneration; wildfires; Landsat 8 OLI; normalized burned ratio index

## 1. Introduction

The severity and impacts, at the regional and global scales, of the burned area resulting from wildfires have increased over the last few decades [1]. Climate change has been identified as an important factor that contributes to these increases [2–5]. Climate change affects wildfires, both directly, through the weather conditions that affect fire ignition and propagation, and indirectly, through its effects on vegetation and fuels [6].

Another type of factor responsible for the increased impacts and severity of wildfires identified by other authors was the socioeconomic factor. For example, the abandonment of crops or agroforestry land, the disinvestment in the forestry sector, and other human practices also contribute to the increase in biomass fuel [7–13].

Wildfires cause many environmental problems because the physical–chemical and biological properties of soil are affected, the aerosols and other particles emitted during combustion cause air pollution, and the pollutants deposited on the surface of burned areas' soils can be transported by water and pollute water bodies [14]. The vegetation, beyond all the benefits it brings to the environment, is of utmost importance for attenuating some

of these environmental problems, in particular the soil erosion and water contamination factors mentioned above.

In general, ecosystems are partially modified when a wildfire occurs, the vegetation is burned partially or totally, and the biological activity is perturbed [15]. The natural regeneration of vegetation after a fire depends on the soil condition (the severity of the fire is important) [16–19], the vegetation species present (growth by the rhizomes or not), the bank of seeds available (for germination in the burned areas) [20], and the weather conditions or climate characteristics.

Wildfires are a big problem in the Mediterranean region due to their social, economic, and environmental consequences [21]. Many landscapes in this territory have been affected by fire in the past, but fire is considered a natural factor with an important influence on the biological productivity and composition of several ecosystems [20]. In fact, fire creates open areas, which favors the germination of species by removing the established vegetation, and has direct effects on germination and seed survival [22]; fire may facilitate the germination and development of several species by changing the mineral environment [23]. Some plant species are resilient to fire, such as *Quercus faginea*, while others are more easily consumed by fire due to their flammable substances, e.g., pine resin. Portuguese forests have experienced major changes in the past few years, mainly because of the conversion to eucalyptus, which can be explained by the increasing demand for this raw material by the cellulose industry. These changes have had an impact on the fire regime of this territory: the events are increasingly intense, occur more frequently, and the extent of the burned area is increasing because of the difficulty of extinguishing the fires [24].

Many Portuguese wildfires are recorded every year, resulting in a large burned area [24]. The severity of fire in this territory is often high [25] because the weather conditions provide excellent conditions for fire propagation due to the particularly high temperatures, low relative humidity, and significant wind speed. Conversely, the high availability of combustible biomass fuel also contributes to fire intensification, and these factors sometimes result in uncontrollable wildfires. The climate in Portugal promotes the occurrence of wildfires, as it has a rainy season during the spring, which is favorable for the development of vegetation, followed by a very warm period that triggers the development of large wildfires [26].

The assessment of vegetation recovery in burned areas is crucial in land management [27]. This assessment is important to prevent or remediate different environment impacts, but vegetation recovery is not always a positive factor in the context of wildfire recurrence because these areas are then particularly susceptible to future wildfires. However, vegetation recovery assessment is complex because it depends on a variety of biological and environmental factors and on the interaction between them [15]. Post-fire regeneration depends mainly on the initial vegetation and on environmental factors—climatic and terrain characteristics—present onsite [28].

Several methodologies have been proposed for vegetation assessment, namely, methodologies based on monitoring the vegetation state from spectral indices [17]; for example, the Normalized Difference Vegetation Index (NDVI) [29–31], the Soil-Adjusted Vegetation Index (SAVI) [32–34], the Leaf Area Index (LAI), the Fractional Vegetation Cover (FVC) [35,36], the Regeneration Index [37–39], the Normalized Difference Infrared Index (NDII) [34,40–42], and Spectral Mixture Analysis (SMA) [43,44]. Other spectral indices, such as the Normalized Burn Ratio (NBR) and Enhanced Vegetation Index (EVI), which combine and extract useful information from several spectral bands [30], have also been widely used to study fire-induced vegetation changes, including burn severity [45] and regeneration dynamics [27,30,32,46,47]. In recent studies, the factors that determine post-fire regeneration patterns were investigated, and this analysis integrated the improvement of the predictive models of vegetation dynamics [39,48–53].

The recurrence of wildfires in Portugal is higher, in particular in the central region [24], where few preventive measures have been adopted in forest areas (for example, the construction of paths through the forest for access by firefighters, the obligation of owners to

clear the forest, etc.). In these areas, the natural growth of vegetation without restrictions and the biomass fuel available can induce new, large wildfires. In this sense, the vegetation recovery assessment assumes great importance for the management of these territories, particularly the creation of preventive and reactive measures to apply in forest areas. This is a premise advocated by the research community—for example, Lentile et al. [54] found that indicators of burn severity, and thus potential ecosystem recovery, could prove useful to post-fire planners tasked with strategically rehabilitating areas likely to recover slowly or in undesirable ways.

The main goal of this research is the assessment of vegetation recovery in burned areas of central Portugal (study area, Figure 1) through remote sensing data, using Landsat 8 OLI images, and biophysical data. The assessment of vegetation recovery was performed differentially by vegetation type (low- or high-growing vegetation and predominant species) and biophysical data (for example: precipitation, temperature, insolation, relief characteristics, etc.).

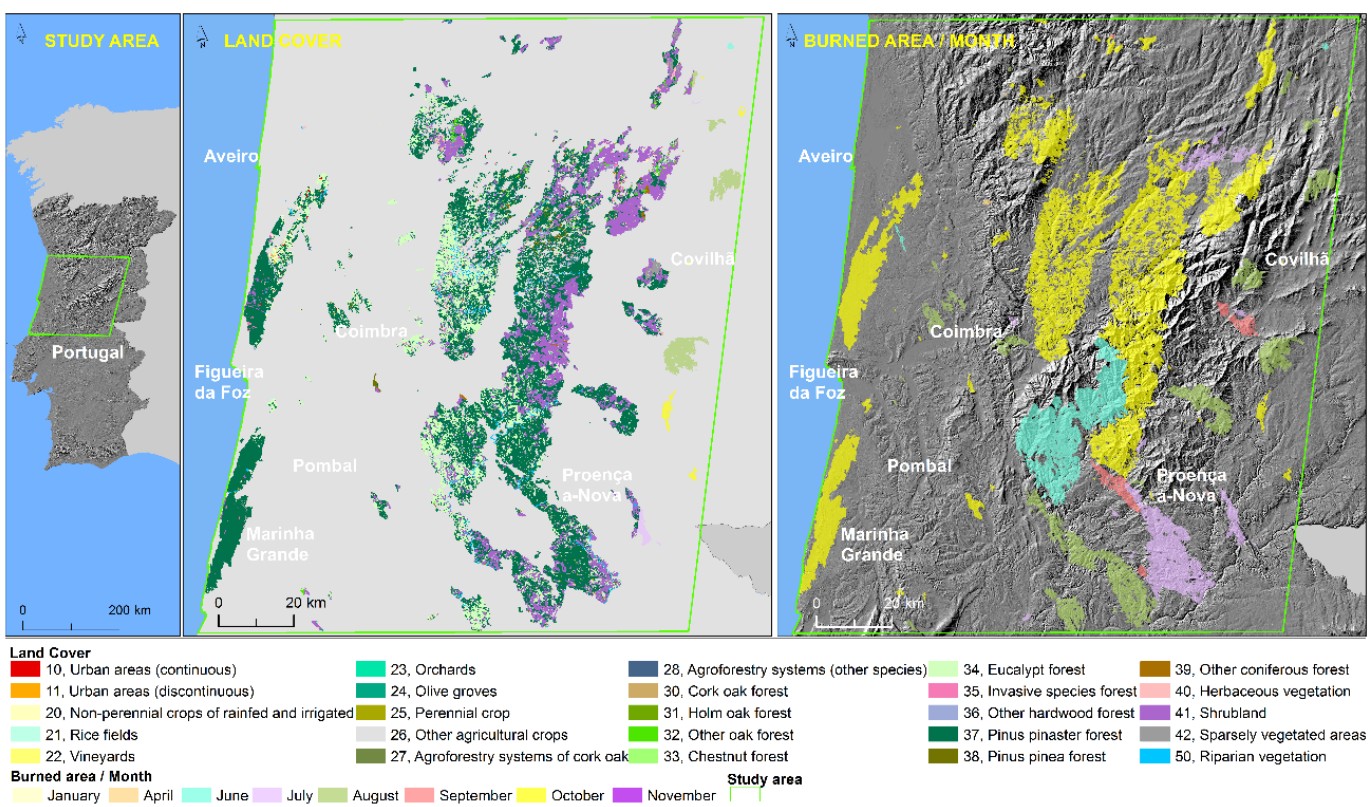

**Figure 1.** Study area (center of Portugal): land cover and burned area by month of 2017.

## 2. Materials and Methods

### 2.1. Study Area

This research was developed in the central region of mainland Portugal (Figure 1), in a 20,867 km$^2$ study area. This selection took into account the wildfires that occurred in this territory in 2017, with dozens of civilian casualties, large material damages (including to the forest), and vast burned areas, with the consequent perturbation of ecosystems.

In the coastal and central areas of the study area, *Pinus pinaster* forest prevails, but eucalyptus forest is common in the central region, while, in areas to the east, shrubland occupies vast areas, resulting essentially from the wildfires that occurred over the last decade [24].

Wildfires occurred in different months throughout 2017, with important occurrences in June and July, but the wildfires of October were the most important ones considering

the extension of the burned area, the number of occurrences at the same time, and the distance between wildfires. In these wildfires, a large area of *Pinus pinaster* was devastated at the coast. This was a very old pine forest, planted in these areas to attenuate erosion and stabilize the soil, in particular through the expansion of sand from the maritime coast. The high-temperature conditions during the fire season and the low precipitation are climate factors that aggravated the occurrence of fires during 2017, making them even more devastating and difficult to control [55].

*2.2. Data*

The data used in this research are described in Table 1. Landsat 8 OLI images were selected instead of wildfire occurrence: a pre-fire image was used for the determination of land cover (LUC) of the previously existing unburned vegetation, in order to compare with the regeneration of vegetation after the fire. For this, we selected images corresponding to six different dates, using only images without clouds.

**Table 1.** Data description.

| Data | Description | Source |
|---|---|---|
| Landsat 8 OLI images | Dates: pre-fire—6 January 2017; post-fire—6 November 2017; 16 June 2018; 8 October 2018; 12 January 2019; 13 February 2019; 4 May 2019 | Landsat USGS Global Visualization Viewer (GloVis) |
| Environmental biophysical | Land Cover Map (COS) | General Directorate for Territorial Development |
| | Elevation (DEM), slope, aspect, curvature | Digital Elevation Model (DEM) from the GMES RDA project (EU-DEM), made available by the European Environment Agency |
| | Soil; soil and subsoil potential permeability (water infiltration capacity); soil thickness; soil pH | EPIC WebGIS Portugal |
| | Annual precipitation; average annual temperature; average annual insolation; simple continentality index | Portuguese Environment Agency and Monteiro-Henriques et al. [56] |
| Burned area | Burned areas and date of wildfires | Institute for the Conservation of Nature and Forest (ICNF) (Portugal) |

Several environmental biophysical variables were previously collected. In the environmental biophysical dataset, the Wald test ($p < 0.05$) was applied and only significant variables were selected for vegetation recovery assessment (Table 1).

Land cover was obtained by supervised classification of the Landsat 8 OLI images, namely the images of 4 May 2017, and 25 LUC classes were defined. This output was validated with control points extracted from the Land Cover Map (COS) available from the General Directorate for Territorial Development (DGT) of Portugal (accuracy > 85%). Field work was performed to validate the vegetation geoinformation obtained by supervised classification and for monitoring recovery after the fire.

For the analysis, some LUC classes presented in Figure 1 were grouped into a new class, "other land cover types," because these classes have a reduced area or are not mostly occupied by vegetation, but were affected by wildfires (e.g., discontinuous urban areas). The shrubland class includes several species present in these areas (e.g., *Ulex europaeus, Genisteae, Erica arborea, Myrica gale, Phillyrea angustifolia, Prunus spinosa, Rhamnus alaternus,* among others) [57].

In the burned areas, vector data available from the Institute for the Conservation of Nature and Forest (ICNF) (Portugal) website were selected for 2017, and only these burned areas were integrated into the modeling process. The total burned area in that year in the study area was 372,625.4 ha. The large wildfires of this year occurred in October, but other important wildfires also occurred in June and July. These occurrences are more frequent during the hotter period because summer conditions (high temperature, low humidity, and wind speed) are associated with forest material availability and are more favorable to the development of large wildfires.

Independent variables used in the vegetation recovery assessment are presented in Figure 2. Relief information was obtained using a Digital Elevation Model (DEM), and, thus, elevation was also integrated into the modeling.

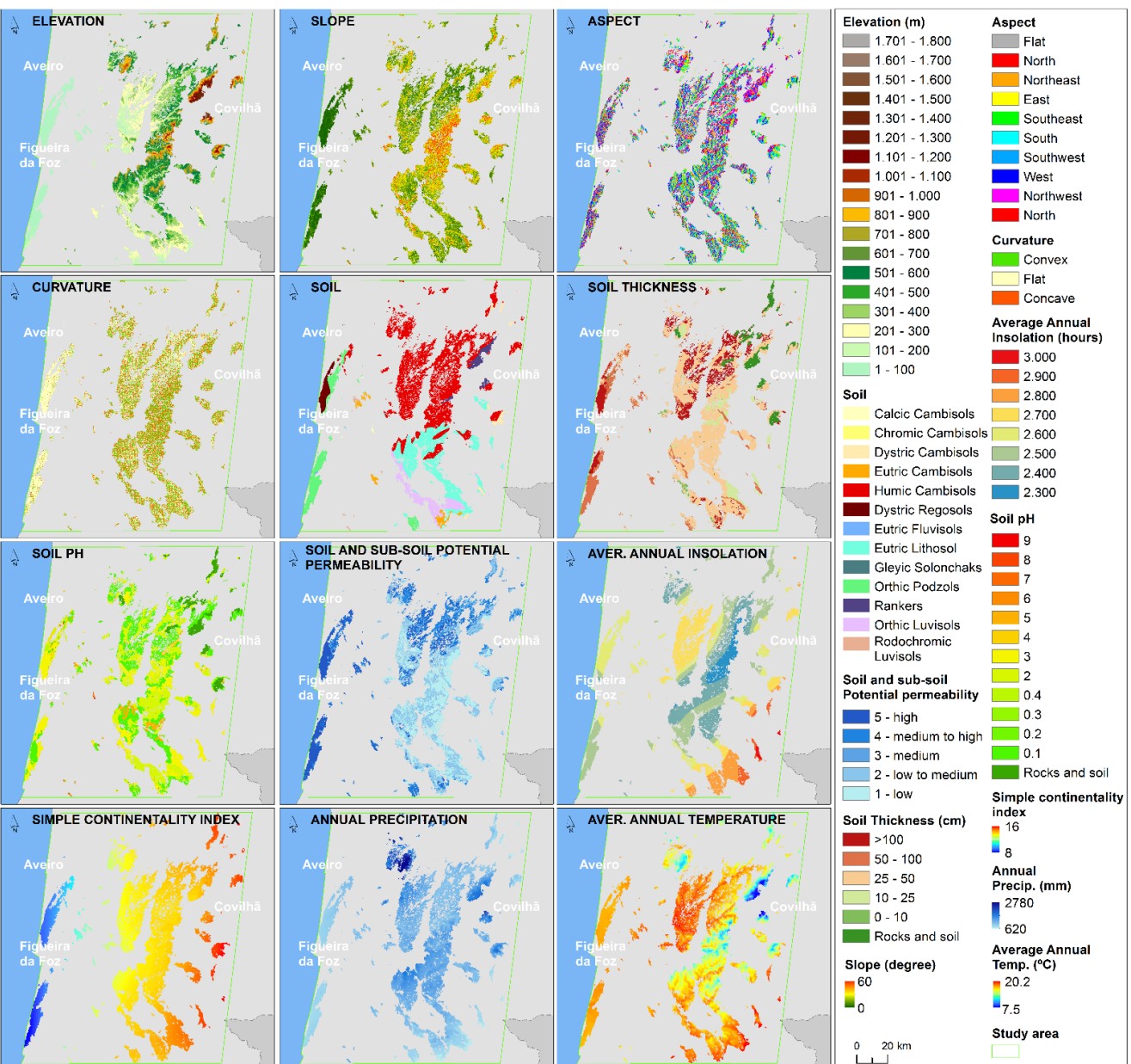

**Figure 2.** Independent variables used in the modeling of vegetation recovery in burned areas resulting from the wildfires of 2017.

The soil properties were considered in the research because those characteristics are important for the fixation of certain vegetation species and their growth [58]. Soil and

subsoil potential permeability represents the qualitative assessment of the water infiltration capacity of the soil, considering the influence of the geological substrate, soil, and slope (the land use parameter was not considered) [59]. Soil pH influences the availability of different elements' absorption by plant roots, such as nutrients and toxic elements, as well as the activity of soil microorganisms [60], and the soil thickness conditions, fertility, and usable water storage capacity, allowing for the determination, to a large extent, of the soil's suitability for the growth of certain species or groups of species [58].

Precipitation is an important driving force influencing the production of different plant communities [61,62], and this variable also was considered in the assessment. The precipitation in the study area is influenced by the relief, which is steeper at higher elevations. Temperature is another variable in the research because it is a primary factor that affects the rate of plant development [63]. The study area presents different temperatures, also influenced by relief; they are lower when the elevation is higher.

In the Mediterranean region, vegetation recovery can be rapid due to the adaptation of vegetation to the disturbance by fire. Some important factors include low competition for sunlight, increased nutrient availability, and reduced water losses by transpiration [64]. The sunlight factor was introduced to the vegetation recovery assessment through insolation data (average annual hours).

The continentality gradient was also important for delimiting the distribution of the forest [65] because land surfaces have lower effective heat capacities as well as usually reduced evaporation rates. The effect of this factor may be moderated by proximity to the ocean, influenced by the direction and strength of the prevailing winds that interfere with temperature, precipitation, and air humidity content. Large burned areas in the study area occurred along the coast, and the vegetation recovery was and will be influenced by the continentality factor. This factor also justifies the higher temperatures for the interior of the territory, having as an implication the reduction of relative humidity, especially during the hottest seasons of the year, leading to more severe fires due to the reduced water content in forest materials.

### 2.3. Methods

The Normalized Burn Ratio index (NBR) was used to highlight burned areas and to estimate the severity of fires (Figure 3). Burned areas have high reflection in shortwave infrared (SWIR) and low reflection in near infrared (NIR) and will, therefore, have a low NBR value. Similar to the Normalized Difference Vegetation Index (NDVI), unburned vegetation will have a high NBR value [47,66]. NBR was calculated with Landsat 8 OLI imagery, using the following equation:

$$NBR = (NIR - SWIR)/(NIR + SWIR) \tag{1}$$

where NBR is the Normalized Burn Ratio index, NIR is the near infrared (Band 5), and SWIR/Thermal is the shortwave infrared (Band 7).

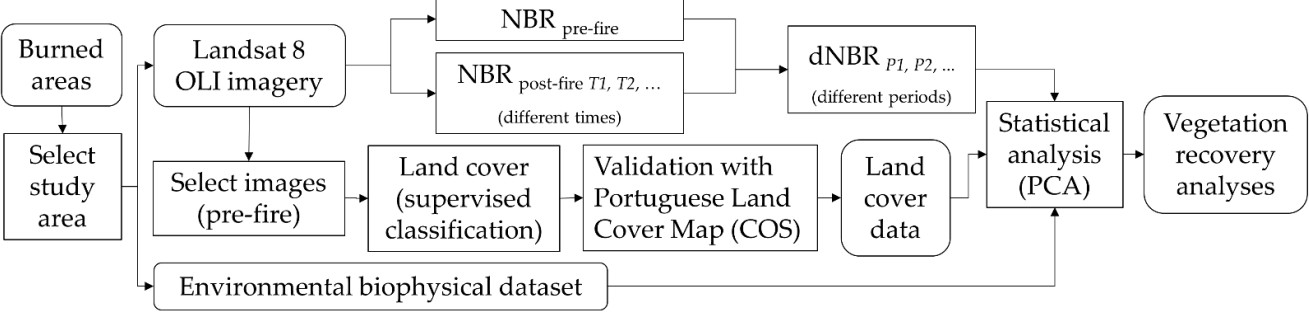

**Figure 3.** Workflow of vegetation recovery analyses in burned areas.

The differenced NBR product (dNBR) was also used to map burn severity [46,66]. In this product, pre-fire and post-fire represent the timing of the acquisition of the imagery [67], and can be obtained by the following equation:

$$\text{dNBR} = (\text{NBR})_{\text{pre-fire}} - (\text{NBR})_{\text{post-fire}} \tag{2}$$

The NBR index and dNBR were obtained using Geographical Information Systems (GIS) for different periods (Table 2). GIS was also used to harmonize all geoinformation used in the research and to perform analyses of vegetation recovery assessment using the selected independent variables. All geoinformation was converted using a raster resolution of 30 m, resulting in 4,310,165 cells in the model.

**Table 2.** dNBR products in different periods (*P*).

| dNBR | Period |
|---|---|
| *P1* | 6 January 2017–6 November 2017 |
| *P2* | 6 January 2017–18 June 2018 |
| *P3* | 6 January 2017–8 October 2018 |
| *P4* | 6 January 2017–12 January 2019 |
| *P5* | 6 January 2017–13 February 2019 |
| *P6* | 6 January 2017–4 May 2019 |

The vegetation shows different levels of growth depending on its biophysical characteristics [15]. Thus, to evaluate the influence of these factors on vegetation recovery in burned areas, the dNBR*P6* was cross-referenced with elevation, slope, aspect and curvature of hillsides, soils (type); soil and subsoil potential permeability; soil thickness; soil pH, annual precipitation; average annual temperature; average annual insolation; and simple continentality index.

All environmental biophysical data have been standardized to integrate the statistical analysis (Figure 3). The dNBR results were used to calculate the correlation between different LUC types and to perform the Principal Component Analysis (a multivariate exploratory technique), to obtain the LUC groupings with similar characteristics in terms of vegetation recovery. These statistical analyses were done using Statistica 7 software (Stat Soft. Inc., Tulsa, OK, USA). A total of 4,310,165 observations/cases were considered in this analysis.

## 3. Results

### 3.1. Burned Area Distribution by LUC

The wildfires of 2017 devastated large forest areas, but many cropland areas were also affected, for example, olive groves or nonperennial crops that are rainfed and irrigated, as shown in Table 3 by cross-referencing burned areas with LUC (pre-fire). However, three LUC types were predominant in burned areas: *Pinus pinaster* forest, eucalyptus forest, and shrubland. These three LUC types accounted for 81.9% of the burned area in the study area in 2017. Other LUC types were also affected by the wildfires (e.g., agricultural crops or urban areas), but are not representative and were grouped under "Other land cover types".

### 3.2. Burned Ratio Index

Wildfires occurred in different months, but large wildfires occurred mainly in the summer and at the beginning of the autumn (a period with reduced seed germination capacity and reduced vegetative growth). The short period between these large wildfires did not allow for the regeneration of vegetation in the burned areas. This was observed during field work, where the growth of vegetation was evaluated.

The severity of fire is spatially diversified; for example, the wildfires of October show areas with high fire severity along with other areas with less severe effects. This

differentiation was observed particularly in the first NBR results after the fires (Figure 4), where the central burned areas presented higher values of NBR.

**Table 3.** Pre-fire land cover areas in burned areas of 2017 (inside the study area).

| Land Cover | Area (ha) | Area (%) |
|---|---|---|
| *Pinus pinaster* forest | 156,726.8 | 42.06 |
| Eucalyptus forest | 93,912.4 | 25.20 |
| Shrubland | 54,400.3 | 14.60 |
| Other hardwood forest | 15,745.8 | 4.23 |
| Other agricultural crops | 13,619.2 | 3.65 |
| Nonperennial crops of rainfed and irrigated | 8375.0 | 2.25 |
| Other oak forest | 5589.1 | 1.50 |
| Olive groves | 4437.1 | 1.19 |
| Invasive species forest | 3910.4 | 1.05 |
| Other land cover types | 15,909.2 | 4.27 |

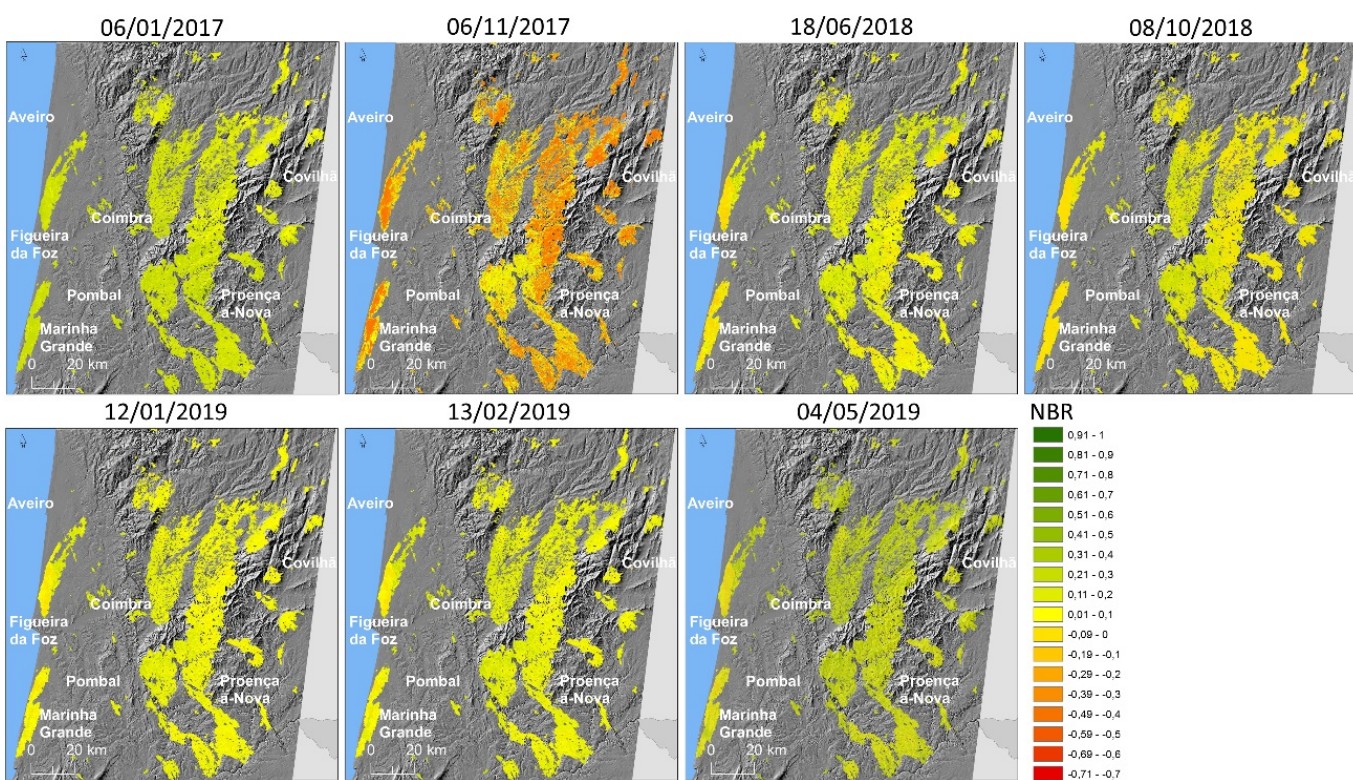

**Figure 4.** Normalized Burned Ratio index (NBR) in burned areas of wildfires of 2017.

### 3.3. Differenced Burned Ratio Index

In a short period after the occurrence of wildfires, it was observed that the vegetation did not recover or recovered very little, as can be seen from the analysis of the dNBR*P1* (Figure 5). The last large wildfires occurred in October 2017 (Figure 1) and the time that elapsed until the end of the period of dNBR*P1* was very reduced; however, besides this conditioning factor, we must also take into account the season (autumn), as the vegetation does not regenerate easily due to the climatic conditions (progressively cooler temperatures and first rainfalls).

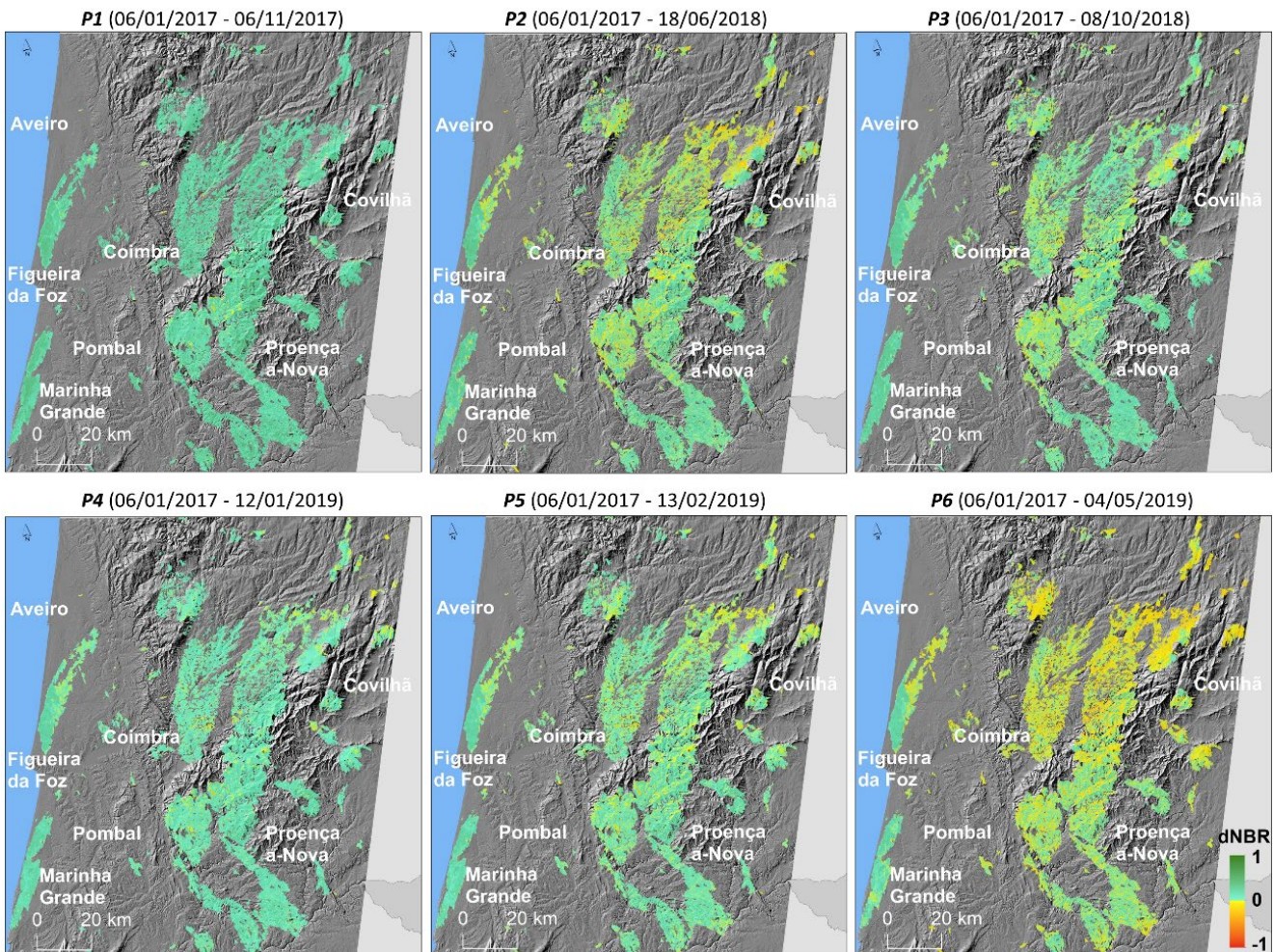

**Figure 5.** Differenced NBR product (dNBR) of burned areas in study area.

In the following year after the fire, particularly after spring, we observed the regeneration of vegetation in large patches of burned areas; this is shown in the dNBR*P2* values, with a greater emphasis on burned areas further north where shrubland predominates. This recovery was slowed by the arrival of autumn (dNBR*P3*); the recovery resumed and was more marked from February 2019 (dNBR*P5*), the year in which the regeneration of much of the vegetation in the burned areas stands out, with the exception of pine forests along the coast (dNBR*P6*).

Analyzing the results of each dNBR, we saw variations concerning the statistical description (Table 4), with the highest average value in the dNBR*P1* because it was the shortest post-fire period. The lowest value occurred in the dNBR*P6*, being negative in this case. These results indicate a greater recovery of the vegetation in this last period, with some areas of more vegetation compared to what occurred before the fire.

**Table 4.** Statistical description of dNBR products in different periods (*P*).

| Statistics | P1 | P2 | P3 | P4 | P5 | P6 |
|---|---|---|---|---|---|---|
| Min. | −0.97 | −0.68 | −0.61 | −0.54 | −0.55 | −0.77 |
| Max. | 0.70 | 0.64 | 0.56 | 0.93 | 1.04 | 0.57 |
| Mean | 0.20 | 0.07 | 0.08 | 0.09 | 0.07 | −0.01 |
| Std. dev. | 0.10 | 0.15 | 0.11 | 0.08 | 0.09 | 0.13 |

The spatiotemporal distribution of dNBR is distinct depending on the post-fire period. The vegetation's regeneration is more pronounced when considering a longer period, hence the dNBR*P6* is not similar to the remaining results of dNBR (Figure 6). In the burned areas in the interior, especially in the north of the study area, the vegetation recovery process is more accentuated (areas with negative dNBR), which means that the post-fire NBR values are higher than those observed pre-fire.

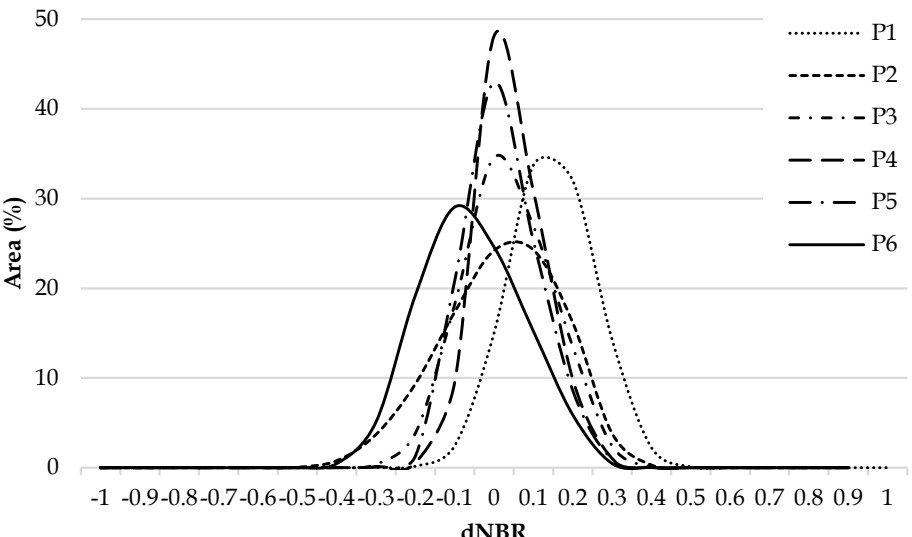

**Figure 6.** Distribution of dNBR product of different periods (*P*).

The results of the dNBR allowed us to distinguish the regeneration of the vegetation according to the season of the year. In this case, with the transition from winter to spring, the vegetation had a greater regeneration capacity, as shown by the transitions from *P4* to *P5* and *P6*.

The longer the period after the fire, the greater the recovery of the vegetation should be. Thus, it is important to analyze in greater detail the dNBR with the longest period—in this case, the dNBR*P6*.

Certain vegetative types have greater recoverability capacity, a fact observed, for example, in herbaceous vegetation and oak forest (Figure 7, negative values). In this recoverability, the phenological processes of the vegetation must also be considered, in particular during the spring season, when the environmental conditions (precipitation and temperature) lead to greater development of most vegetation.

In the areas of *Pinus pinaster* forest, eucalyptus forest, and shrubland, the vegetation present smallest recovery periods. In these cases, the vegetation is essentially arboreal and shrub, and the recovery process differs depending on the size of the vegetation: faster in the shrubs vegetation where the vegetal composition is a differentiating factor (including this class several species that naturally recover after fire, without human intervention), while arboreal vegetation needs much more time to recover.

The recovery also differs according to the forest species present, *Pinus pinaster* takes longer to regenerate compared to eucalyptus if there are conditions for the regeneration process of *Pinus pinaster* to occur. When the recurrence of wildfires is high in the pine forests, the availability of seeds for regeneration is compromised because the pine-trees do not have enough time to grow and produce new seeds.

Certain LUC types show a behavior very similar in terms of dNBR, and this is highlighted in the correlation coefficients of dNBR presented in Table 5. For example, shrubland presents high correlation with herbaceous vegetation, and eucalypt forest with *Pinus pinaster* forest.

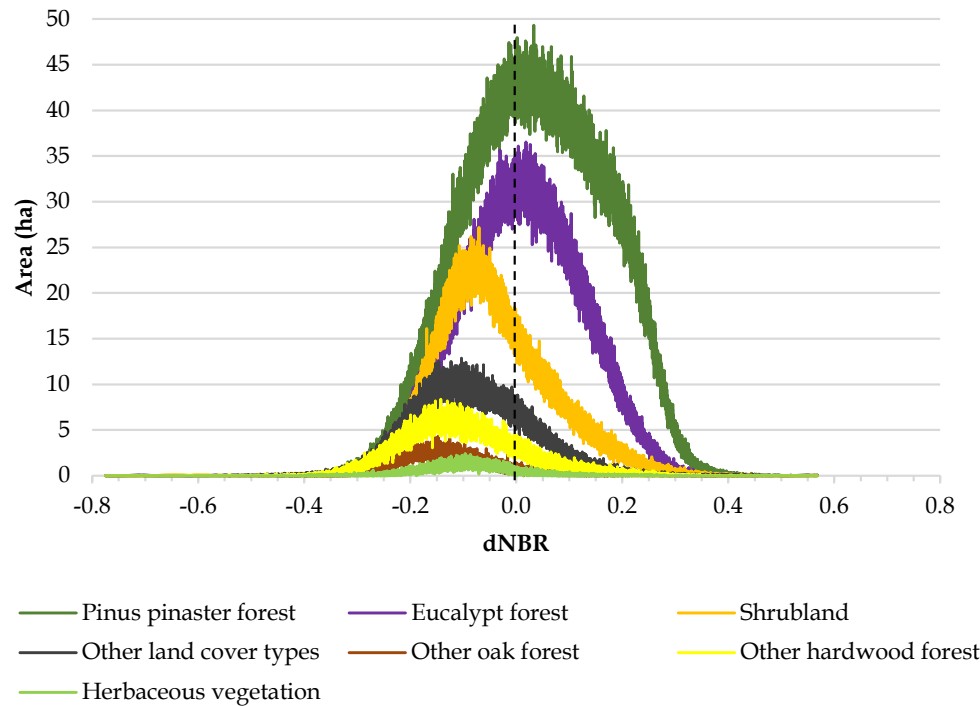

**Figure 7.** dNBR*P6* distribution in the main LUC classes.

The results of principal component analyses show different groupings of LUC classes of dNBR, specifically forest species with a minor area of representation (in absolute terms), for example forests of cork oak, holm oak, chestnut, *Pinus pinea*, and other conifers. Associated with these LUC classes, other LUC classes are affected by wildfires, in particular the crop land cover types (olive groves and agroforestry). LUC classes with more area affected by wildfires tend to group in the PCA at higher linkage distances (Figure 8). These results reflect the different regeneration capacity in time of the different species (*Pinus pinaster*, eucalyptus, and shrubland).

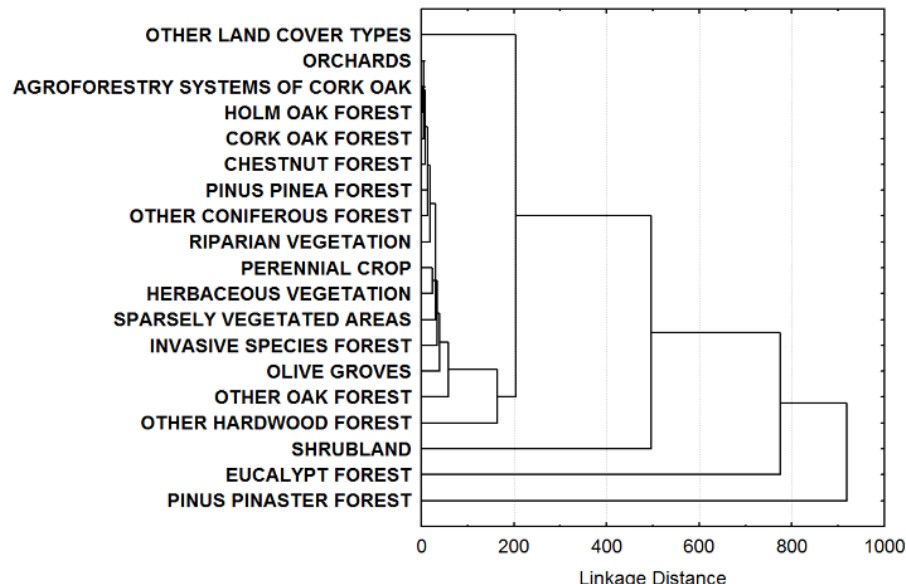

**Figure 8.** Euclidean distances (single linkage) of dNBR*P6* of LUC classes.

**Table 5.** Correlation coefficients between dNBR*P6* of LUC classes (significance level $p < 0.05$).

| | Other Land Cover Types | Olive Groves | Perennial Crop | Other Oak Forest | Eucalypt Forest | Other Hardwood Forest | *Pinus pinaster* Forest | Herbaceous Vegetation | Shrubland |
|---|---|---|---|---|---|---|---|---|---|
| **Other land cover types** | **1** | **0.92** | **0.89** | **0.86** | 0.66 | **0.94** | 0.54 | **0.88** | **0.93** |
| **Orchards** | 0.49 | 0.43 | 0.43 | 0.49 | 0.25 | 0.5 | 0.18 | 0.43 | 0.41 |
| **Olive groves** | **0.92** | **1** | 0.84 | 0.74 | 0.71 | 0.85 | 0.6 | **0.85** | **0.93** |
| **Perennial crop** | **0.89** | 0.84 | **1** | 0.79 | 0.59 | **0.86** | 0.48 | 0.82 | 0.85 |
| **Agroforestry systems of cork oak** | 0.32 | 0.31 | 0.3 | 0.27 | 0.21 | 0.31 | 0.17 | 0.32 | 0.33 |
| **Cork oak forest** | 0.51 | 0.5 | 0.46 | 0.43 | 0.35 | 0.49 | 0.29 | 0.48 | 0.51 |
| **Holm oak forest** | 0.17 | 0.17 | 0.14 | 0.13 | 0.1 | 0.15 | 0.09 | 0.18 | 0.18 |
| **Other oak forest** | **0.86** | 0.74 | 0.79 | **1** | 0.32 | **0.92** | 0.22 | 0.78 | 0.7 |
| **Chestnut forest** | 0.57 | 0.48 | 0.52 | 0.63 | 0.2 | 0.62 | 0.12 | 0.51 | 0.45 |
| **Eucalyptus forest** | 0.66 | 0.71 | 0.59 | 0.32 | **1** | 0.51 | **0.95** | 0.55 | 0.82 |
| **Invasive species forest** | 0.72 | 0.72 | 0.66 | 0.53 | 0.75 | 0.64 | 0.76 | 0.65 | 0.79 |
| **Other hardwood forest** | **0.94** | 0.85 | **0.86** | **0.92** | 0.51 | **1** | 0.39 | **0.85** | 0.83 |

**Table 5.** *Cont.*

| | Other Land Cover Types | Olive Groves | Perennial Crop | Other Oak Forest | Eucalypt Forest | Other Hardwood Forest | *Pinus pinaster* Forest | Herbaceous Vegetation | Shrubland |
|---|---|---|---|---|---|---|---|---|---|
| *Pinus pinaster* forest | 0.54 | 0.6 | 0.48 | 0.22 | **0.95** | 0.39 | **1** | 0.44 | 0.72 |
| *Pinus pinea* forest | 0.71 | 0.74 | 0.64 | 0.47 | 0.73 | 0.61 | 0.63 | 0.66 | 0.79 |
| Other coniferous forest | 0.69 | 0.69 | 0.64 | 0.52 | 0.66 | 0.62 | 0.59 | 0.64 | 0.75 |
| Herbaceous vegetation | **0.88** | **0.85** | 0.82 | 0.78 | 0.55 | **0.85** | 0.44 | **1** | **0.88** |
| Shrubland | **0.93** | **0.93** | 0.85 | 0.7 | 0.82 | 0.83 | 0.72 | **0.88** | **1** |
| Sparsely vegetated areas | 0.64 | 0.72 | 0.56 | 0.29 | 0.81 | 0.48 | 0.69 | 0.58 | 0.77 |
| Riparian vegetation | 0.55 | 0.6 | 0.48 | 0.31 | 0.59 | 0.44 | 0.49 | 0.49 | 0.6 |

### 3.4. The Importance of Biophysical Factors on Vegetation Recovery of Burned Areas

The most important factors affecting the vegetation recovery in burned areas are climate factors and soil characteristics. This is shown by the positive correlation of the analyses with dNBR*P6* (Table 6).

**Table 6.** Correlation coefficients between dNBR*P6* and environmental biophysical variables (significance level: $p < 0.05$).

|  | dNBR*P6* |
| --- | --- |
| Temperature | 0.47 |
| Soil | 0.45 |
| pH soil | 0.45 |
| Continentality | 0.45 |
| Precipitation | 0.41 |
| Insolation | 0.31 |
| Soil thickness | 0.25 |
| Aspect | 0.24 |
| Curvature | 0.24 |
| Elevation | 0.21 |
| Slope | 0.19 |
| Permeability | 0.14 |

Temperature is the most important independent variable in vegetation recovery, with a higher positive correlation value between environmental biophysical variables and dNBR*P6*. These results, analyzed in detail, indicate that lower temperatures influence the vegetation recovery; this is more evident in higher areas, where the temperatures are in general lower compared to areas with lower elevation.

Precipitation is also important in the vegetation recovery process, and this factor is also influenced by the topography of the study area. Burned areas located in the interior and to the north of the study area have more precipitation compared to the rest of the study area. Both latitude and relief also have an important influence on the recovery process.

The continentality gradient is also important in this process. Indirectly, this factor reflects the influence of the prevailing winds from the north and from the Atlantic Ocean that interfere with temperature, precipitation, and air humidity content, all of which are important factors in the germination of seeds and plant growth.

The soil type and some physicochemical characteristics are also important factors. Vegetation recovery is most efficient in humic cambisols and in soils with reduced pH.

In contrast, topographic factors present lower correlation values, although positive, indicating that they also influence the vegetation recovery in burned areas.

The aspect of hillsides is the most important topographic factor on vegetation recovery in burned areas. Hillsides exposed to the north and northwest have more efficient vegetative growth due to the exposure of these hillsides to precipitation and to masses of air rich in humidity that prevail in these directions; for hillsides facing south, insolation is a more important factor and can explain the low humidity in the soils and air, which interfere with vegetation growth.

The elevation has a lower positive correlation. The burned areas with high elevation do not present high regeneration of vegetation post-fire because these areas have low temperatures and are more exposed to the wind (which interferes with the growth of plants). It was also observed that, in areas with low elevation, sheltered from the winds, and with mild temperatures, the regeneration of vegetation is more efficient.

## 4. Discussion

In the burned areas, the analyzed vegetation recovered more from wildfires that occurred between June and August. The fire severity in these burned areas was lower relative to those areas affected by the wildfires of October. However, the recovery process of burned vegetation was more pronounced here, first, as a result of the species present pre-fire that regenerate relatively fast post-fire (e.g., eucalyptus and shrubland—*Genisteae*), and second, due to the time between the date of the wildfire and the date of the first Landsat 8 Oli image (post-fire), with more regeneration of vegetation in the areas burned by the first wildfires, relative to those areas affected by the wildfires of October.

Some studies found that regeneration of vegetation occurs within the first two years following fires [68–70]. This fact was observed in this research, where some areas have more vegetation relative to the pre-fire situation (especially shrubland), i.e., in these cases, it took approximately 19 months to regenerate all the vegetation. Cerdà and Doerr [64] found that post-fire vegetation recovery in the Mediterranean territory can be rather rapid due to the adaptation of vegetation to disturbance by fire and, following burning, the low competition for sunlight, nutrient availability, and reduced water losses by transpiration. Other post-fire forest floor conditions such as the amount of bare soil exposed are also important determinants of post-fire vegetation recovery [54].

The vegetation recovery dynamics in burned areas is different according to the species present before the fire and the vegetation size (trees, herbs, and shrub). Eucalyptus resprouts readily after fire [71], so eucalyptus forest regenerates relatively quickly, and this growth was observed in the NBR results. However, in areas with this vegetation type, other vegetal species are also present, for example herbaceous or shrub species. The mixed vegetation is important regarding fire severity, especially when species with lower flammability are present [72,73]. The burned areas analyzed were composed of diversified vegetal species (before the fire), although there are predominant species (e.g., *Pinus pinaster* or eucalyptus), also present are the species with lower flammability mentioned above (*Quercus rotundifolia*, *Q. robur*, *Q. faginea*, *Q. pyrenaica*, and *Q. suber*), which contributed to the spatial differentiation of fire severity (which is reduced) [74], a factor that also influences the vegetation recovery process [17,30,39,47]. Although some species of *Quercus* can be flammable, they are well adapted to fire. For example, *Quercus suber L.* is the only Mediterranean species that can regenerate from the crown, even in the case of severe fires, allowing it to recover more quickly than other species. In forests of *Quercus faginea* (a minimally flammable species), few trees die due to fires (about 5%) [75].

NBR results provide a means to differentiate the severity of wildfires [46], with emphasis on the fire severity that affected the burned areas resulting from the October wildfires. The weather conditions were the primary factor, according to reports from the Portuguese civil protection services, for the high fire severity observed, specifically the high temperatures, wind speed, and low relative humidity (ranges for these variables are available in the report of IPMA [76]).

The negative values of the last dNBR observed in the northern part of the study area indicate more regeneration of vegetation relative to other burned areas. The fire severity in these areas was high, but the vegetation present pre-fire was essentially shrubland, composed of species that regenerate easily after fire (e.g., *Ulex* and *Genisteae*), thus explaining the results obtained. In this sense, we highlight the importance of vegetation resprouting in the burned areas because resprouting provides persistence under disturbance [77] and enables more efficient recovery of the vegetation land cover [54,78].

In contrast, the ashes deposited in the soil surface after a fire provide nutrients that are available to the plants, with this ash representing a substantial part of the nutrient stock [79]. This factor could also have contributed to the efficiency of vegetation regeneration, but seeds and/or vegetal species that grow from rhizomes must be present. However, losses of nutrients by ash convection (conversion of elements to solid, inorganic ash subject to wind action) increase with fire intensity [80].

Nutrients resulting from the burning of forest material are available in burned areas if they are not lost by direct volatilization or ash convection [80]. The ash deposited on the surface of soils (a process in which low-intensity fire has an important role) constitutes a factor to provide nutrients to the plants and the sustainability of the forest [81]. In this process, the amount of precipitation that falls on the burned areas is an important factor to take into account. When the water retention capacity of the soils is exceeded, surface runoff occurs and these nutrients are carried away by the water [14,82]. This factor can affect the recovery of vegetation on the burned areas resulting from wildfires, especially those that occurred in October, because rainfall in the following months is high. In the burned areas resulting from the wildfires of May, June, and July, some plants had already germinated and contributed to the reduction of these losses.

The influence of the seasons on the vegetation recovery is very clear when analyzing the dNBR results. The biggest vegetative growth occurs mainly during the spring, characterized by the mildest temperatures and the availability of water (in soil and in terms of air moisture), factors that allow for the development of plants [83–85]. The correlation obtained between dNBR and environmental biophysical variables point to the importance of climate characteristics for vegetation recovery, especially temperature and precipitation. These results are in accordance with other studies that examine the influence of these variables [28,43,50,51]. Additionally, the continentality gradient plays an important role in the recovery process because it also indirectly reflects the spatial distribution of precipitation and temperature.

A limitation of using NBR to calculate the vegetation recovery index has to do with the identification of those areas where some forest species are predominant and to distinguish them from areas covered with other vegetation with fast growth rates after fire. For example, eucalyptus and *Pinus pinaster* forests recover at different times, as observed in the results, but the first vegetation appearing after the fire in this forest areas is herbaceous; this vegetation type is also considered during the NBR calculus.

The soil type is also important for vegetation recovery. The results point to the importance of this variable in the recovery processes, and, when analyzed in detail, the most important relationship was observed between humic cambisols and the vegetation recovery process. This soil type has an umbric A horizon, which is thicker and rich in organic matter—favorable factors for plant recovery. However, the soil properties can be affected by fire severity [86,87].

Soil $p$ also has a positive correlation with vegetation recovery. The pH inevitably increases due to soil heating as a result of organic acid denaturation, with significant increases at high temperatures (>450–500 °C) [88]. This factor favors vegetation recovery and can explain the results obtained.

Topographic factors will have different impacts on the regeneration of different forest cover types [17,39]. Effectively, the results show that relief characteristics influence the vegetation recovery, but they are not the most important factor in the recovery process. Several studies found an influence of slope in this recovery; for example, Shakesby [89] found that steeper slopes tend to retain less water and be more susceptible to soil erosion and nutrient loss, which explains why most soils in these slopes are thinner and less productive. Cerdà and Doerr [64] observed differential recovery rates on north- and south-facing slopes.

The results of this research are based on the assumption of the importance of spatial planning actions in forest areas affected by wildfires. Usually, the planning actions do not take into consideration the potential regeneration of the vegetation in burned areas, but assume, at least in some cases, that the type of LUC existing pre-fire continues unchanged. Thus, the actions resulting from planning have to be reconsidered, especially in the strategical rehabilitation of burned areas [54]. The previous assumption is used, for example, in the creation of the official LUC maps of Portugal. The burned areas identified during cartographic creation do not present the same vegetative regeneration index as shown in

this research; in some areas, the forest does not regenerate because the species present before the fire, when burned, do not recover partially or completely (e.g., *Pinus pinaster*).

## 5. Conclusions

The vegetation recovery process does not occur in the same way in the burned areas of the study area. This fact points to the interference of different factors, such as the severity of fire, the vegetation species, and the environmental biophysical characteristics of these areas.

With the computation of dNBR from the NBR product at different times (obtained by Landsat 8 OLI images), the dynamics of the post-fire vegetation recovery were assessed; different spaciotemporal dynamics were observed and different patterns were found. The recovery was more marked in some burned areas, especially in burned areas located in the interior and to the north of the study area, but was also influenced by the season of the year.

The vegetation species present in the burned areas pre-fire are important in the process of vegetative recovery because some species regenerate more quickly compared to other species—for example, eucalyptus regenerates in a short time compared to coniferous, oak, or invasive species, and some species also regenerate by the rhizomes.

Climate conditions (temperature and precipitation), continentality gradient, and the properties of soils (type and pH) are great influences on the vegetation recovery of the burned areas assessed. Topographic factors also influence this recovery, but are not the most important variables in this process.

The results of this research are important to spatial planning actions, in particular for forest planning: first, in the recovery of ecosystems; second, to prevent the occurrence of new large wildfires by adopting new strategies of forestry and the modification of anthropogenic practices (e.g., reforestation with native species resistant to fire—for example, *Quercus faginea*, *Quercus ilex*, and *Quercus rotundifolia*; changes in crop activities that use fire to clean the fields, among others). Thus, we stress the importance of the recurrence of fire to be evaluated, especially in areas with certain plant species that are sensitive to fire and species that do not recover integrally (e.g., a high recurrence of fires in pine forests does not allow for the growth of these species until they start producing seeds).

**Author Contributions:** All work presented in this manuscript was prepared by the author. All authors have read and agreed to the published version of the manuscript.

**Funding:** This work was financed by national funds through the FCT-Portuguese Foundation for Science and Technology, I.P., under the framework of the Research Unit UID/GEO/00295/2013 (Centre for Geographical Studies and Associated Laboratory TERRA, Institute of Geography and Spatial Planning, Universidade de Lisboa).

**Informed Consent Statement:** Informed consent was obtained from all subjects involved in the study.

**Data Availability Statement:** Data are available and can be shared upon reasonable request.

**Acknowledgments:** The author is grateful to his colleague Ana Silva for reviewing the text.

**Conflicts of Interest:** The author declares no conflict of interest.

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
