# Peer review of "Vegetation Recovery Patterns in Burned Areas Assessed with Landsat 8 OLI Imagery and Environmental Biophysical Data"

_fire, doi:10.3390/fire4040076_

Round 1

Reviewer 1 Report

I find the topic of the manuscript entitled 'Vegetation recovery patterns in burned areas assessed with Landsat 8 OLI imagery and biophysical data' of interest for a publication in MDPI Fire. I found the state of the art exhaustive and detailed, the introduction section is well presenting the paper topic and previous work in literature and the discussion gives a detailed description of ecological processes, occurring during post-fire vegetation recovery. Methods section is not clear, sometimes also confusing study area and literature review. I suggest to include a workflow figure to help the reader understand the analysis steps and how the many datasets have been used. The manuscript is well written from the standpoint of the English.
I consider the adopted experimental design not appropriate, since it do not take under consideration variability related to vegetation phenology. In fact, results shown in maps demonstrate that dNBR values decrease after summer 2018 and increase again after winter 2019. NBR spectral index has been used to identify changes from post-fire, even years after the fire events occurred, that is a novelty compared to previous research studies which generally make use of vegetation spectral indices. Nevertheless, pre-fire date corresponds to a winter acquisition, when photosynthetic activity is reduced especially in deciduous forests, and depending on the presence of understory. I believe a comparison of pre-fire and post-fire conditions during vegetation seasonal peak (e.g. spring) could make the results more consistent. In fact, the results obtained by the comparison named 'P4' seems more consistent (winter pre-fire VS winter post-fire) than those obtained from a comparison of different year periods. LANDSAT scene used for Land Cover classification (acquired during May 2017) could in example be used as NBR reference for pre-fire vegetation conditions. Even though field works are mentioned, the study lacks of validation procedure related to vegetation recovery.
Nevertheless, I believe that the author is presenting detailed description of ecological processes occurring during post-fire vegetation recovery, and I would suggest to revise the experimental design in order to compare pre- and post-fire conditions at the same phenological phase.
In conclusion, I would suggest to reconsider the manuscript after major revision.

I have specific comments and suggestions in order to improve paper readability:
Line 25: I warmly suggest a final sentence to specify how the research study results contributed to address future research studies and how managers could benefit from the outcomes.
Line 48: I suggest to consider also fire severity in the list.
Lines 83-85: I suggest to include the following references, with the analysis of full biophysical index LAI, not mentioned among the other indices, to monitor post-fire recovery: (1)Filipponi, F.; Manfron, G. Observing Post-Fire Vegetation Regeneration Dynamics Exploiting High-Resolution Sentinel-2 Data. Proceedings 2019, 18, 10. https://doi.org/10.3390/ECRS-3-06200. (2)Han, A.; Qing, S.; Bao, Y.; Na, L.; Bao, Y.; Liu, X.; Zhang, J.; Wang, C. Short-Term Effects of Fire Severity on Vegetation Based on Sentinel-2 Satellite Data. Sustainability 2021, 13, 432. https://doi.org/10.3390/su13010432.
Line 102: I would suggest to provide here some example of biophysical data.
Lines 120-121: It is not clear how burned area were obtained. If not retrieved from other repositories (e.g. EFFIS), have the LANDSAT data been used for the burned areas identification. How fires occurred in the various months have been identified? Please clarify.
Line 125: From my point of view, variables can be related to (bio)climatic, geophysical (soil properties) and geomorphological characteristics. Since the term biophysical is often used to define physical measure of biological properties, I suggest to clearly define the definition of biophysical data used in the manuscript. I expect it is more related to enviromental biophysical definition, considering its inclusion in the manuscript title.
Line 140: Has aspect been considered for the modelling using values as degrees (e.g. from North)? Generally speaking it is better to use directional components to account for orientation instead of aspect in degrees.
Line 141-148: This part is more related to the description of the study area rather than data.
Line 179: From the maps there are soil pH values much lower than 3.5 (ultra acid soils), which looks strange. I suggest to check the correctness of values reported in the specific map legend.
Lines 216-219: These sentence looks like study area description, and could be moved to study area or discussion sections.
Line 220: I would stress that these results comes from the pre-fire land cover classification done in the research study.
Lines 232-233: Here field works are mentioned. Nevertheless, it has not been specified if field campaigns have been used for validating the results nor the ny methodology to use the field information described. Please clarify.
Line 271: There are missing Lines in Figure 5 plot legend.
Line 280-281: Could this result for deciduous forest be related to the vegetation phenology, considering that post-fire acquisition is in May month?
Line 299: In order to improve readibility, I suggest a correlation matrix plot with both numbers and colors in place of Table 5.
Lines 351-352: I am wondering why the generation of a post-fire land cover map has been considered, in order to compare pre-fire and post fire ecosystems. I believe a strong assumption has been done about persistent vegetation type cover, even after the wildfire, while grasslands and shrubs may replace forests during the initial vegetation recovery process.
Line 367: Here again a strong assumption is done, that is immediately eucalyptus forest grows where eucalyptus forest was. Since climax status achievement could be reached after tens of years after a disturbance event, has plant succession be considered?
Lines 460-461: It is not clear if you refer to time of the year when fire occurred or time of the year for post-fire acquisition considered to evaluate the recovery.

Author Response

Dear Reviewer 1,

thanks for the comments to the manuscript. They are very elaborate and detailed and will undoubtedly help me to enhance its content. I would also like to make some comments with the purpose explaining better my options. Please check my detailed response in the attached file.

Reviewer 2 Report

Great collection of data-rich figures and interesting exploration of two-year vegetative recovery following several 2017 Portuguese wildfires.  My suggestions revolve around adding more context to the study, including vegetation and fire regimes as well as more details on the wildfires, more clearly differentiating types of recovery (whether it is of the pre-fire community or other species), and greater detail and examples of the implications of the study for managing for the next wildfires.  I also have suggestions for the figures and tables and offer a few editorial suggestions (also see attached manuscript). Here are the major issues, and these and other issues are detailed below:

  • Please add more context on the major vegetation types, their major species, and response to fire as well as what is known about the fire regime across the region before and since widespread human settlement. Also add greater detail on the wildfires.
  • Your study encompasses both burn severity, as evidenced the season following burning, as well as 2-yr post-fire vegetation recovery. Since you are using remotely sensed images, vegetation recovery may consist of the pre-burn species, or conversely contributed by species different than those in the pre-burn community.  The results need to clearly differentiate these different classes of vegetation recovery.  You allude to these different types of recovery but it should be highlighted.  For example, the speed of recovery and high correlation between the dNBRP6 of Pinus pilaster and Eucalypt forests simply reflect similar levels of vegetation and not a return of the pre-burn vegetation type.  Keeley (2009. Fire intensity, fire severity and burn severity: a brief review and suggested usage. International Journal of Wildland Fire 18(1) 116-126 https://doi.org/10.1071/WF07049) provides a good discussion of fire severity and resource impact.
  • Given the rich database that you put together, can you provide more specific suggestions for planning for the next fire? What vegetation types were most severely impacted? What soil types or temperature zones have vegetation types likely to burn severely and not recover well?  It might be helpful to focus in on some examples to help the reader better understand your suggestions.  For example, you mentioned the need to plant native, fire-resilient species.  Can you give us an example of a non-native vegetation community that burned severely and did not recover well? 
  1. 46 (and throughout): Ecosystems are not necessarily destroyed by fire; fire burns through forests and shrublands, but rarely “destroys” them. Similarly, on lines 47-48, biological activity is potentially altered by fire, but rarely destroyed. And, on line 49, delete the clause “once it determines the degree of destruction caused.” On line 110, instead of “destruction of ecosystems,” consider saying “alteration of ecosystems.”

Paragraph beginning on line 53: Consider adding more ecological context here.  It would be helpful to provide an overview of major vegetation types and their response to fire.  For example, Pinus pinaster is a native tree that is fairly resilient to surface fires, but is killed by crown fires and relies on nearby seed-producing trees to regenerate (is that correct?).  Eucalypt spp. are also native (is that right?) and sprout vigorously when top-killed by fire.  The shrublands are apparently composed of a number of native fire resilient species that sprout vigorously.  What is known about the historical fire regime in the various vegetation types? And how has fire occurrence changed over time?

l. 91: what preventive measures do you mean?

l.113-118: It would be useful if you provided more information, perhaps in an appendix, about each fire, including the dates, size, ignition source, vegetation types, and weather conditions.

l. 173-178: Please add more explanation of the “continentality gradient”. That is, the scale ranged from 8 to 16, where eight represents “X” and 16 represents “Y.” Values between 8 and 16 are “Z”.

Fig. 2 (and other figures).  Beautiful figures!  Please check that the colors in the legend match the figures.  For example, the curvature legend shows three colors: green, blue, and red; yet along the coast I see a light yellow.

Fig. 5: check the legend. I see 6 lines, but the legend only includes 4. Also consider switching to colors; it is hard to decipher the different lines.

Fig. 6: Increase the line width in the legend and double check that the colors match the figure.  For example, the lines for Pinus pinaster forest and Eucalypt forest are so thin it is hard to tell what color they are; same with other hardwood forest.

l. 284: the finding that Pinus pinaster forest and Eucalypt forest have equally short recovery periods is a bit counter-intuitive, since Pinus pinaster does not sprout, right? This section needs to be rewritten to clarify that the apparent quick recover of Pinus pinaster forests is not due to pine recovering (right?), but rather to the recovery of other vegetation.

l. 288: explain what you mean by “naturally recover.”

Table 5: Consider reducing this table to just the types that had significant correlations, and state that the other types were not significantly correlated with any other types. 

Fig. 7: Consider omitting this figure; it doesn’t add much information for the space it encompasses.

Table 6: Consider reorganizing from the most significant variables to the least significant variables.

Discussion:

l. 356, 360 (and elsewhere): consider using “found” instead of “refer”

l. 373: are these oak species less flammable or just strong sprouters? Or maybe both?

One of the potential values of the paper is offering post-fire insights into future forest management, which the last paragraph briefly touches on.  Although not clearly presented as a result of the study, you suggest the need to reforest with “autochthonous” (native?) species, resistant to fire, and changes in crop activities that use fire to clean the fields.  What native species do you recommend? And did your study provide evidence of native species being more resilient to the fires? And, can you add more details about results that suggested burning crop fields led to problems?  Were some of the fires the result of these field fires getting out of control?  The last sentence of the conclusion is hard to decipher, but it sounds like the recommendation is to identify ecosystem types composed of species that are particularly sensitive to fire and recover slowly if at all.  And if this interpretation is correct, are you suggesting that mitigating factors should be taken in these fire-sensitive systems to prevent widespread burning?  What vegetation types are you referring to and what specifically would you recommend?

Author Response

Dear Reviewer 2,

thanks for the comments to the manuscript. They are very elaborate and detailed and will undoubtedly help me to enhance its content. I would also like to make some comments with the purpose explaining better my options.

Round 2

Reviewer 1 Report

I find the topic of the manuscript entitled 'Vegetation recovery patterns in burned areas assessed with Landsat 8 OLI imagery and biophysical data' of interest for a publication in MDPI Fire. The manuscript describes the results of a numerical exercise using spectral indexes for the identification of burnt areas and the data assimilation in numerical modeling to estimate the erosion process.
I consider the new re-submitted version of the manuscript much improved with respect to the previous version.
I appreciated the introduction of a workflow showing of the different processing steps and dataset used. Despite the author defended some of the statements in the manuscript with the description of the process and pictures in the author's reply to reviewers (very appreciated), the experimental design has not been revised in order to compare pre- and post-fire conditions at the same phenological phase. Nevertheless, the results obtained with the proposed methodology support the discussion, that gives a detailed description of ecological processes, occurring during post-fire vegetation recovery.

Author Response

Dear reviewer 1,

I really appreciate your praise for the work presented, I did everything considering your excellent opinion on the various points raised by your comments.

Regarding what is missing (compare pre- and post-fire conditions at the same phenological phase), this work would be interesting to present here in this manuscript, but it would imply substantial changes in the base, which would imply changing the entire article. However, more simplified information on this matter has already been added in the round 1 review, so I haven't changed it in this new round. I also inform you that this comparative work has been carried out and will be presented in a new manuscript to be submitted soon to the Fire Journal, where the recurrence of forest fires until this summer will also be evaluated.

Once again, thank you very much for contributing and substantially improving this manuscript.

Reviewer 2 Report

I looked at the revised paper as well as the response to my previous comments.  It appears that you addressed some of my suggestions in the manuscript, but it is hard to tell. I can’t link your responses to any of my specific comments.  For example, I’m not sure which of my comments your statement “A workflow has been added” refers to. And it appears that in some cases, my suggestions are addressed in the responses, but not by making changes in the manuscript.  Such as your reasoning for which images you chose; you should add this explanation to the manuscript as well.  And other times, the response is not adequate.  For example, if you refer to a website, please give the url for that website in the manuscript.  Also, I see you added a little explanation about the “continentality gradient” in the paper, but you did not explain how you assigned the values; what does an “8” mean? A “16”?  I am attaching my original comments and requesting that you copy my comments into a document and then add your response below each of my suggestions, indicating if you agree and what changes you made in the paper.  And if you don’t agree, please provide your reasoning for not changing the manuscript.  As an example, I suggested that you delete figure 7; I did not find your response to that suggestion and you did not delete that figure; please tell me why.       

Author Response

Dear Reviewer 2,

thanks for the new comments to the manuscript. I would also like to make some comments with the purpose explaining better my options.

I would mention that in a review phase where there are multiple reviewers, it is necessary to combine different suggestions and sometimes there may even be contradictions between the comments of each reviewer. It is up to the author to make options for a better review, agreeing or not, but as long as it is scientifically substantiated.

Round 3

Reviewer 2 Report

Thank you for the detailed response to my comments—very helpful indeed.  You have addressed many of the issues I raised in my previous review.  I do understand that many of your changes were in response to other reviewer comments.  That’s why it is very helpful if you include all suggestions from other reviewers and your responses as well.  That way reviewers can better understand your reasoning for changes. 

I have no major issues, but I do still have a question about the oak species in your study area.  You mention on lines 63-64 and again on lines 402-403 that the oak species are resilient to fire because their leaves are less flammable.  And, on lines 504-505 you recommend planting oaks because they are resistant to fire.  Unless I missed it, you didn’t mention that the oak species are also resilient to fire because they sprout vigorously when top-killed. I am not familiar with these particular species of Quercus, but many (most?) other Quercus species are strong sprouters.

Interesting paper, and the additional context you provided really enhanced it.  The only other suggestion I have is that you have the paper reviewed by a native English speaker.  There are lots of awkward phrases and odd words that detract from an otherwise very interesting story.

Author Response

Dear reviewer,

I send the comments in the attached file.

Thanks for everything.
